# Role of Reactive Astrocytes and Microglia: Wnt/β-Catenin Signaling in Neuroprotection and Repair in Parkinson’s Disease

**DOI:** 10.3390/ijms262411880

**Published:** 2025-12-09

**Authors:** Margherita Grasso, Chiara Mascali, Francesca L’Episcopo

**Affiliations:** OASI Research Institute—IRCCS, Via Conte Ruggero 73, 94018 Troina, Italy; mgrasso@oasi.en.it (M.G.); cmascali@oasi.en.it (C.M.)

**Keywords:** Parkinson’s disease, astrocytes, microglial cells, inflammation, neurodegeneration, neuroprotection, neurogenesis

## Abstract

Parkinson’s disease (PD) is a neurodegenerative pathology defined by specific, distinctive signs, primarily the progressive loss of dopaminergic neurons (DAergic) in the substantia nigra pars compacta (SNpc), associated with gliosis phenomena. The mechanisms that trigger the degeneration of DAergic neurons are not yet fully elucidated, although it is recognized that the interaction between genetic and environmental factors acts as a critical modulator of neuronal vulnerability. Strong evidence points to glial reactivity as a central element in PD pathophysiology; however, it remains a controversial topic whether this activation has a protective effect or, on the contrary, whether it contributes to exacerbating DAergic neuronal loss. The use of MPTP (1-methyl-4-phenyl-1,2,3,6-tetrahydropyridine)—a neurotoxic substance—represented a turning point in Parkinson’s research, allowing the clarification of various molecular mechanisms of the disease. The primary aim of this review is to explore the current state of knowledge regarding the role of astrocytes in the processes of DAergic neurodegeneration, neuroprotection, and neurorepair. We focused on the relationship between astrocytic origin factors and neurogenic signals that mediate MPTP-induced plasticity in DAergic neurons of the nigrostriatal system. The contribution of reactive astrocytes in promoting DAergic neurogenesis starting from Neural Stem/Progenitor Cells (NPCs) present in the adult midbrain is also analyzed. Among the mediators released by astrocytes, we have previously identified the Wnt/β-catenin signaling pathway as a fundamental element capable of positively influencing neuroplasticity and dopaminergic neuronal repair induced by the toxic MPTP. In conclusion, deciphering the intrinsic plasticity of nigrostriatal DAergic neurons and signals that facilitate communication between astrocytes and NPCs is crucial for the identification of potential therapeutic targets aimed at stimulating neuronal repair.

## 1. Introduction

Parkinson’s disease (PD) is positioned as the second most widespread neurodegenerative disease, immediately after Alzheimer’s, with a prevalence of approximately 1% in the elderly population, due to a pathological accumulation of Alpha-synuclein (α-syn) protein in oligomer or fibril forms [1]. The main hallmark of the disease is the selective loss of dopaminergic (DAergic) neurons in the substantia nigra pars compacta (SNpc), leading to a reduction in dopamine levels that underlies the characteristic PD motor symptoms. Most cases of PD are diagnosed when symptoms are already advanced, but evidence suggests that neuronal damage begins much earlier. Clinical manifestations, such as stiffness, akinesia, and tremor at rest, occur only when there is a significant loss of mesencephalic dopaminergic neurons in the central nervous system (CNS) suggesting the role of several mechanisms able to compensate for the neuronal damage [2,3]. The drugs used for the treatment of the disease, such as dopaminergic agonists, anticholinergic drugs, or monoamine oxidase inhibitors, can mitigate motor and/or non-motor symptoms and, therefore, improve the quality of life of patients only in the early stages, but do not slow down the progression of the disease and its devastating effects [4]. Less than 10% of Parkinson’s cases are of genetic origin; everything else appears to be sporadic and probably represents an interaction between genetic and environmental factors [4,5,6]. The most affected gender is male, but estrogen deficiency due to menopause and aging in women is certainly considered a risk factor [7,8]. In genes involved in dopamine (DA) metabolism, inflammatory processes, and mitochondrial functions, several polymorphisms have in fact been reported [9], and it has also been proposed that environmental factors, such as contact with pesticides and heavy metals or infectious agents in the first years of life, can increase the risk of manifesting the disease. Conversely, the consumption of caffeine and tobacco, a healthy and balanced diet, movement, and social relationships seems to reduce the risk [9]. Numerous epidemiological studies have demonstrated an inverse association between caffeine intake and the risk of PD development in both men and women, although this protective effect appears to be more pronounced in males [10]. Caffeine, a non-selective antagonist of adenosine A1 and A2A receptors abundantly expressed in the striatum, is involved in the modulation of dopaminergic activity. Caffeine antagonism of these receptors may enhance dopaminergic transmission and offer neuroprotection [11]. Furthermore, caffeine and other compounds present in coffee have antioxidant properties and can modulate inflammation, contributing to neuroprotection, and also influence the gut microbiota and motility, factors that are increasingly linked to the pathogenesis of PD [12]. Despite solid epidemiological evidence, randomized clinical trials on the use of caffeine as a neuroprotective agent are limited; however, the results are not always definitive. Most of the evidence comes from observational studies, which cannot establish causality.

Tobacco smoking has an even stronger and more consistent association with a reduced risk of PD than caffeine, a finding that initially surprised researchers, since the negative impact of smoking on overall health is known. Indeed, cohort and case-control studies have shown that current and ex-smokers have a significantly lower risk of developing PD than non-smokers [13]. Nicotine, the major tobacco alkaloid, has been shown to act on specific nAChRs present on dopaminergic neurons. Activation of these receptors can stimulate DA release and modulate neuronal function, potentially offering neuroprotective effects [14].

Although smoking is generally pro-inflammatory, some studies suggest that nicotine may have anti-inflammatory effects in the brain at low doses or in specific settings, modulating microglial activity [15]. Tobacco smoke contains compounds that inhibit the activity of MAO-A and MAO-B in the brain. MAO-B inhibition is the basis of drugs approved for PD (e.g., selegiline, rasagiline), suggesting a plausible pharmacological mechanism for protection [16]. The exact nature of the “protective factor” in tobacco is still a matter of research, and it is unclear whether it is nicotine itself or other compounds. In conclusion, while both caffeine and tobacco are associated with a reduced risk of PD, as demonstrated in epidemiological studies, the underlying mechanisms are complex and not fully elucidated. The scientific research aimed to explore these compounds in order to identify specific active ingredients and signaling pathways that can be employed for the development of novel neuroprotective therapeutic strategies for PD, without the harmful effects associated with tobacco use.

As in the experimentally induced Parkinson’s model, it has been seen that some nonsteroidal anti-inflammatory drugs (NSAIDs) can reduce the risk and severity of the disease [17]. Genetic factors, interacting with events that occurred during the first years of life, can also predispose individuals to the disease [18] (Figure 1). Evidence on how much environmental factors and neuroinflammation are involved in the pathogenesis of Parkinson’s disease emerged when an individual developed parkinsonian syndrome following accidental injection of MPTP (1-methyl-4-phenyl-1,2,3,6–tetrahydropyridine) [18]. Postmortem analysis highlighted reactive microglia aggregates in the vicinity of neurons, suggesting an active neurodegenerative process that was perpetuated for years after initial toxic exposure. This has led to the hypothesis that this condition could be supported by a chronic neuroinflammatory condition. Subsequently, numerous postmortem and experimental epidemiological studies have indicated that inflammatory mechanisms associated with glial cell activation, and modulated by pro- and anti-inflammatory factors, contribute to the physiology of PD [19,20]. In the neurodegenerative process, however, the role of inflammation remains a matter of debate, as the causal reaction of these two events has not been fully demonstrated.

It is known that glial cells exert a key role in neurodegenerative diseases, including PD. From one perspective, glial cells, primarily astrocytes and microglia, have a crucial role in brain homeostasis, immune response, and neuronal survival; on the other hand, in PD, astrocytes can contribute to both neuroprotection and neurodegeneration, as well. Astrocyte gliosis and microglial cell activation contribute to the neurodegenerative process, increasing the dopaminergic neuronal loss [21]. Inflammation can also exert a neuroprotective role, thus configuring itself as a dual-role mechanism with significant therapeutic implications for neurodegenerative diseases [9,17,22,23,24]. In fact, the inflammatory microenvironment can modulate adult neurogenesis in a deleterious or favorable sense, based on glial activation and the specific characteristics of each brain region [25]. Numerous studies indicate that glial cells play a much more important role in CNS health and disease than previously recognized. Brain development, neurotransmission, neuronal survival and differentiation, inflammatory and neuroprotective pathways, maintenance of the blood–brain barrier function, and neurogenesis depend primarily on glial activity. The identification and/or development of new therapeutic strategies aimed at both aspects of the glial response could lead to a greater understanding of the mechanisms of the disease and could be crucial in the development of new therapies, both for Parkinson’s and for the other neurodegenerative diseases characterized by glial dysregulation [26].

## 2. Chronic Inflammation as a Determining Factor in the Progression of Parkinsonian Neurodegeneration

Astrocytes and microglia are “activated” in most CNS pathologies, including inflammatory, infectious, ischemic, and neurodegenerative diseases, such as PD [20,22,23]. Activated glia may induce benefit to the host by producing cytotoxic molecules that kill pathogens, virally infected cells, or tumor cells, but they may also be detrimental by killing host cells, particularly neurons. Under physiological conditions, both astrocytes and microglial cells play a key role in neurotransmission and synaptic homeostasis. Once activated, microglia display conspicuous functional plasticity, altering their functional state in response to local stimuli polarizing from a “resting” (or “quiescent”) state to “active” microglia state (M1 and M2 phenotypes) [27], and transforming into a macrophage-like phenotype that involves morphological changes, proliferation, increased expression of cell surface receptors, and the production of neurotrophic and neurotoxic factors to promote neuronal repair and survival or to counteract the neuronal damage, respectively [28]. Astrocytes respond to injury by hyperplasia and hypertrophy of cell bodies and cell processes, and increased expression of the major astrocytic cytoskeletal protein, glial fibrillary acidic protein (GFAP) levels and other molecules, such as S100, inducible nitric oxide synthase (iNOS), and nuclear factor kB (NFkB) [29]. Reactive astrocytes express a wide range of receptors (for cytokines, chemokines, toll-like receptors, and growth factors), thus acting as dynamic modulators of both pro-inflammatory and anti-inflammatory responses [30]. Among the cytotoxic molecules produced by activated microglia, the resident innate immune cells in the CNS, nitric oxide (NO) from inducible nitric oxide synthase (iNOS), and superoxide from the plasma membrane NADPH oxidase (PHOX), represent two key harmful mediators [31]. iNOS is not normally expressed, but it is induced as a part of the activation state in microglia by cytokines [particularly interferon gamma (IFN-γ), tumor necrosis factor alpha (TNF-α), or interleukin-1 beta (IL-1β)], bacterial cell wall components [particularly lipopolysaccharide (LPS)], and oxidative stress. Of particular mention, if iNOS and NADPH oxidase are active at the same time, then microglia might produce peroxynitrite (ONOO-), a potent toxin, which might promote nitration of various proteins, including tyrosine, and produce hydroxyl radicals [32]. Hence, the generation of the free radical NO followed by the production of peroxynitrite may be implicated in neuronal cell death (Figure 2). Accumulation of ROS, NO, prostaglandins (PGs), and pro-inflammatory cytokines (including TNF-α, IL-1β, and IFN-γ) in the nervous systems of PD patients further supported that a state of chronic inflammation characterizes PD brain [33]. The first evidence of neuroinflammation in PD is supported by postmortem studies that reported an upregulation of major histocompatibility complex (MHC) molecules in the substantia nigra [19]. Subsequent studies then revealed increased levels of β2-microglobulin, ROS, NO, cytokines, and complement components in affected brain regions [34,35]. Chronic neuroinflammation induces neuronal damage and death through several mechanisms, including oxidative stress response, astrocytes and microglia activation, and increased pro-inflammatory cytokine release, leading to the activation of the apoptosis process and loss of dopaminergic neurons [36], supporting the hypothesis that chronic inflammation is a hallmark of PD pathophysiology. In agreement with the inflammation hypothesis, epidemiological analysis has indicated that nonsteroidal anti-inflammatory drugs (NSAIDs) may prevent or delay the progression of PD, acting as neuroprotective agents by inhibition of cyclooxygenase (COX) enzyme and cytokine release, showing an anti-inflammatory activity in a dose-dependent manner [17]. However, the prolonged use of this class of drugs is often associated with significant side effects at the gastrointestinal and renal levels, in addition to the appearance of cardiovascular events, factors that limit its clinical use in chronic conditions. In this regard, it is relevant to mention nitric oxide (NO)—NSAID HCT1026 [2-fluoro-a-methyl (1,1′-biphenyl)-4-acetic-4–(nitrooxy) butylester], NO-flurbiprofen, a derivative of conventional NSAIDs capable of releasing nitric oxide, which maintains anti-inflammatory efficacy and significantly reduces side effects. Moreover, this compound has been shown to attenuate the degeneration of DAergic neurons in mouse models of Parkinson’s [37]. Activation of astrocytes has also been seen to be an important element of neuroinflammation in Parkinson’s disease. In postmortem studies, a significant increase in GFAP-positive cells has been seen [38]. Chronic astrocyte activation, induced by inflammatory signals from microglia and neurons, contributes to neurodegeneration in Parkinson’s, but the molecular basis of astrocyte–neuron dialogue still remains unclear [39]. Because astrocytes play a key role in maintaining neuronal homeostasis, metabolic support, and protection from oxidative stress, altering their dialogue with neurons can accelerate disease progression and hinder repair mechanisms. Postmortem studies in Parkinsonian brains have shown a reduction in astrocyte density in the CNS compared to less affected regions, such as the ventral tegmental area and the A8 group [40]. Hence, an alteration and/or reduced efficiency of the astrocyte function to a highly activated microglial phenotype might represent a critical vulnerability factor compromising DAergic neuron self-repair ability [41], based on the dual role of astrocytes that, when are activated, are able to secrete neurotoxic factors that cause the death of neurons and oligodendrocytes, but on the other hand, astrocytes upregulate many neurotrophic factors, which are considered neuroprotective in PD [26,36].

## 3. Glial Mechanisms in Dopaminergic Neuroprotection: A Target Analysis on Astrocytes

Targeting astrocyte-specific protective mechanisms could be considered as a potential therapeutic strategy to counteract DAergic neurodegeneration, and as a cellular target to protect neurons by modulating neuronal function and survival [42]. The critical protective role of astrogliosis in response to acute CNS injury was highlighted in one study where it was demonstrated that selective ablation of reactive astrocytes led to increased neuronal and oligodendrocyte death, increased inflammatory infiltration, impaired blood–brain barrier (BBB) recovery, and more profound functional deficits [43]. Astrocytes have been considered passive support elements of neurons in the central nervous system for many years. However, in recent decades, this point of view has been profoundly revised due to the evidence that such cells express a wide range of receptors for neurotransmitters, neuropeptides, growth factors, cytokines, and toxins [44,45]. Astrocytes play a crucial role in maintaining synaptic homeostasis by removing excess glutamate (Glu) through specific high-affinity transporters located almost exclusively on these cells [46]. Similarly, gamma–aminobutyric acid (GABA) is also eliminated from the synaptic cleft of astrocytes and partially recycled via the coded GABA shunt [47]. Furthermore, astrocytes form an extensive network of extracellular communication, connected not only to other astrocytes but also to oligodendrocytes and ependymal cells through gap junctions [48]. Astrocytes, as well as neurons, are involved in neuronal network excitability, transmission, axonal conduction, and long-term or short-term synaptic plasticity via several mechanisms, including the release of gliotransmitters or local ion homeostasis, regulating the synaptic homeostasis both by spatial K+ buffering and the glutamine–glutamate–GABA shuttle [49]. Astrocytes act as central regulators of neuro–glial communication, modulating synaptic transmission and maintaining CNS homeostasis. It is, therefore, not surprising that their function undergoes profound adaptations in the presence of brain lesions [50].

Under physiological conditions, astrocytes play an essential role in protection against oxidative stress, a function particularly relevant for dopaminergic neurons, cells particularly vulnerable to the oxidation processes of dopamine (DA) that generate hydrogen peroxide (H_2_O_2_), as well as to oxidative metabolism of DA itself and to the interactions between Fe and H_2_O_2_, which through the Fenton reaction, lead to the formation of highly toxic free radicals [51]. In addition, these cells preserve neurons from energy depletion by releasing lactate derived from their own glycogen stores by triggering the so-called astrocyte–neuron lactate shuttle [52,53]. Compared to neurons, astrocytes possess higher concentrations of antioxidant molecules, such as vitamin E, ascorbate, superoxide dismutase (SOD), and glutathione (GSH), which give them a greater ability to counteract oxidative damage. In particular, GSH represents a key element in the detoxification of H_2_O_2_; its release protects neurons from reactive oxygen species (ROS) and nitrogen species (RNS) [54]. This function takes on particular importance for the dopaminergic neurons of the substantia nigra because H_2_O_2_, produced by DA conversion by monoamine oxidase-B (MAO-B) during its auto-oxidation, is generated mainly in astrocytes [55]. In that regard, it is important to emphasize that aging, the main risk factor for Parkinson’s disease, is associated with an increase in MAO-B activity. Experimental models that induce genetic upregulation of MAO-B in astrocytes reproduce the increase in age-related enzymatic activity and result in progressive and selective loss of substantia nigra dopaminergic neurons [56]. Another key antioxidant mechanism involves GSH efflux from astrocytes, mediated by the ATP-dependent transporter, multidrug resistance-associated protein 1 (MRP1). The antioxidant response element (ARE) constitutes a key mechanism of astrocyte neuroprotection, with higher basal and induced levels than neurons. Oxidative stress stimulates glutathione synthesis, increases the localization of the export [57] pump Mrp1e, and activates the transcription factor Nrf2, which translocates into the nucleus and binds to the ARE, promoting the expression of antioxidant and detoxification genes, those involved in lactate transport between astrocytes and neurons, and cholesterol synthesis [58,59]. The astrocytes play a key neuroprotective role in the nigrostriatal dopaminergic system, including the increased release of antioxidants, such as GSH, and neurotrophic factors counteracting oxidative stress, particularly relevant in PD pathogenesis [56,60].

Hormones involved in the stress and reproductive axes interact closely with astroglial cells. Astrocytes, macrophages, and microglia represent fundamental targets of steroid hormones, as they express specific receptors, such as those of glucocorticoids (GRs). These cells act both as producers and recipients of cytokines, growth factors, and neurotrophic molecules in the central nervous system [25,61]. In glial tissues, glucocorticoids additionally regulate the expression of several proteins, including GFAP [62]. Glucocorticoids (GCs) are known for their ability to effectively inhibit the production of nitric oxide (NO) derived from the enzyme iNOS in activated glial cells [63]. These molecules, the main endogenous anti-inflammatory mediators, can therefore play a protective role, where the increase of cytokines in the central nervous system leads to potentially harmful effects. In mouse models lacking the GR receptor, the reduction in the activity of endogenous anti-inflammatory pathways significantly increases the vulnerability of nigrostriatal dopaminergic neurons to cell death. Moreover, in the same animal model, it has been demonstrated that an impairment of endogenous anti-inflammatory pathways significantly increases the vulnerability of nigrostriatal dopaminergic neurons to death induced by neurotoxic factors, suggesting that GC-bound GRs are key susceptibility factors in experimental Parkinsonism models [64]. Gender and steroid profile also influence vulnerability to Parkinson’s disease: epidemiological studies reported a greater incidence and prevalence in men than in women [7]. These data are supported by experimental evidence showing how the nigrostriatal dopaminergic system is modulated by estradiol (E2) in rodents and non-human primates, and that E2 exerts neuroprotective effects in different Parkinson’s models [65,66]. Along with classical genomic and non-genomic mechanisms, E2 also exerts neuroprotective effects via glial cells. In fact, neurons and glia express enzymes for the synthesis and metabolism of steroids, such as P450-aromatase, present in glial cells, which converts androgens into estrogens [66,67]. In mice with early P450-aromatase deficiency, and therefore E2 deficiency in the ventral midbrain, the nigrostriatal dopaminergic system shows greater vulnerability to experimental Parkinsonism, indicating a critical role of E2 for dopaminergic neurons [68]. E2, via ER-alpha and/or ER-beta, reduces macrophage and microglia activation in vitro, limiting the production of pro-inflammatory cytokines, NO, COX-2, PGE2, and MMP-9, in part by modulating NF-kB [69,70]. Astrocyte and microglia responses also vary between genders in experimental models of Parkinson’s [71], and endogenous E2 levels affect both microglial activation and survival of dopaminergic neurons [71]. At the same time, E2 protects astrocytes from neurotoxin-induced damage and promotes their survival and expression of dopaminergic neurotrophic factors, contributing to neuroprotection [71,72,73]. The vulnerability of dopaminergic neurons is influenced by multiple factors (including gonadal hormones and stress, as well as pathogens or endotoxins), which exert their effects by modulating the immune responses of astrocytes and microglia [26,74] (Figure 3).

## 4. The Role of the Glial Pathways in Endogenous Dopaminergic Neuronal Restoration

The intrinsic ability of nigrostriatal DAergic neurons to recover spontaneously after MPTP-induced damage, as well as the loss of their age-dependent self-repair ability, could suggest that among the multiple mechanisms, those intimately associated with glial responses could represent important factors involved in the remodeling of the compromised nigrostriatal environment and/or in the promotion, as well as in the improvement, of endogenous protection and repair mechanisms in DAergic neurons [75,76]. Astrocytes are known to express and release numerous growth factors in in vitro studies, including neurotrophins such as NGF, BDNF, CTNF, and NT-3; glia-derived factors such as GDNF and MANF; general growth factors, such as FGF2 and HGF; and activity-dependent neurotrophic factors (ADNs) [77,78,79,80]. In this context, the increased release of BDNF by astrocytes supports the survival of DAergic neurons, particularly in the midbrain region, or the promotion of brain repair by increasing FGF, indicating their role in neuron survival. In addition, astrocytes can secrete M2-2 glutathione transferase through exosomes, enhancing the growth of DAergic neurons and promoting regeneration [42]. Astrocytes produce matrix metalloproteinases (MPPs), which are essential for extracellular matrix remodeling, and support the migration of neuronal precursors during development, and are able to act as precursors in the adult CNS [81,82]. In the ventral midbrain (VM), astrocytes are critical for the development and survival of dopaminergic neurons, expressing members of the Wnt family (Wnt1 and Wnt5a) and DA-specific transcription factors, such as Pax-2, En-1, and Otx-2. They also promote the proliferation and differentiation of dopaminergic progenitors into mature neurons [83,84]. In particular, the Wnt signaling, a family of highly conserved secreted glycoproteins, has been demonstrated to be essential both in the developmental and disease-related processes.

Wnt glycoproteins bind to several membrane receptors, including members of the Frizzled family of seven transmembrane receptors and the co-receptors low-density lipoprotein receptor-related proteins (LRP5, LRP6), with their subsequent phosphorylation and the recruitment of signal transducers DVL and AXIN to the Wnt-bound receptors. Therefore, in the canonical signaling pathway, a cascade of intracellular reactions is triggered, leading to the inhibition of glycogen synthase kinase-3beta (GSK3β) activity, which blocks β-catenin phosphorylation and degradation, leading to an increased stability of cytosolic β-catenin and its translocation into the nucleus where it activates Wnt target genes by interacting with transcription factors in the TCF (T-cell factor)/LEF (lymphoid enhancer factor) family. The canonical Wnt pathway is negatively modulated by Dickkopf-1 (Dkk-1), a secreted protein that interacts with LRP5/6 and with the transmembrane protein Kremen-2, thus inducing the rapid internalization of the co-receptor LRP6.

In the absence of the Wnt signal, cytosolic β-catenin is phosphorylated by kinases (CK1α, GSK3β) and scaffolding proteins (AXIN, APC), leading to β-catenin ubiquitylation and proteasomal degradation, with consequently low levels in the nucleus and, thus, transcriptional repressors prevent activation of Wnt target genes [84,85,86].

Specifically, the Wnt signaling pathway is a fundamental cellular mechanism in both developmental and adult tissues, as well as governing critical processes, such as neuronal survival and differentiation, axonal development, synaptic plasticity, and neurogenesis [87,88]. In the canonical Wnt pathway, also known as the Wnt/β-catenin pathway, following the binding of ligands (Wnt2, Wnt3, Wnt3a, and Wnt8a) to Frz and LRP5/6 receptors, the degradation of β-catenin is inhibited and, following its accumulation in the cytoplasm and its translocation into the nucleus, the interaction with the TCF-1/complex LEF-1 primarily controls target gene expression involved in cell proliferation, survival, differentiation, and migration [89]. The Wnt/β-catenin pathway is crucial for the generation of dopaminergic neurons in the ventral midbrain (VM), while its role in the adult midbrain, normal or affected by PD, is unclear [90]. In the MPTP mouse model, analysis of 92 mRNAs in the VM showed persistent positive regulation of the canonical Wnt1 agonist, confirmed by in situ hybridization and Western blot [91]. Activated astrocytes in the VM have been identified as potential mediators of Wnt1 signaling, and activation of this pathway has been proposed as a key element in dopaminergic recovery through MPTP-induced nigrostriatal plasticity [92]. In summary, activation of the Wnt1/β-catenin pathway is essential for maintaining a normal number of TH+ neurons in the adult midbrain [93,94].

DAergic neurons in the midbrain show a marked vulnerability to oxidative stress and are particularly sensitive to the reduction of growth factors [3,95]. These, produced by astrocytes, protect neurons from pro-apoptotic stimuli, such as serum deprivation, 6-OHDA, and MPP+ [92,96,97]. In vitro studies on astrocyte–neuron crosstalk have shown that Wnt1 expression in adult astrocytes, increased after MPTP injury, can act as a self-protective compensatory signal [90,98]. In this context, several neurotoxic insults seem to activate a self-defensive pathway in astrocyte–neuron co-cultures, resulting in β-catenin stabilization in dopaminergic neurons [99]. β-catenin thus plays a key role in the defense against oxidative stress and can also act as a transcriptional co-activator for several nuclear receptors involved in the development, maintenance, and neuroprotection of TH neurons [100]. Thus, activation of Wnt1/β-catenin appears to be an interesting pathway that could work in concert with astrocyte-derived factors to maintain integrity and protect TH+ neurons [95,101]. Wnt1-mediated neuroprotection is closely linked to astrocyte response to oxidative stress and post-injury inflammation. It requires stabilization of the β-catenin, which transmits pro-survival signals to the nucleus. This suggests that modulating the Wnt1/β-catenin astroglial pathway may promote the balance between apoptosis and cell survival, supporting neuro-rescue in animal models of Parkinson’s [102]. Astrocytic β-catenin signaling via TCF7L2, a key transcriptional effector of the Wnt/β-catenin pathway, has also been shown to regulate synapse development [103], and it has been hypothesized that Wnt microglia–astrocyte crosstalk is activated by downstream CX3CL1/CX3CR1 signaling, which acts as a critical mechanism for synapse remodeling under physiological and/or pathological conditions [104]. Therefore, an in-depth understanding of the molecular pathways and their crosstalk underlying midbrain neuroprotection will be critical to identify new pharmacological strategies and cell replacement therapies in neurodegenerative disorders, including Parkinson’s disease.

Adult neurogenesis, the generation of new functional neurons from progenitor cells, occurs in specific microenvironments and includes proliferation, neuronal fate determination, maturation, and integration into existing circuits [105,106]. Neural stem cells (NSCs), multipotent and capable of self-renewal, can generate both neuronal and glial lines [107]. In the adult midbrain, neurogenesis remains controversial: there is currently no conclusive evidence on new dopaminergic neurons in the substantia nigra after injury [103,108,109]. Adult neurogenesis is regulated by growth and neurotrophic factors, hormones, neuropeptides, inflammatory mediators, and neurotransmitters. Recent evidence indicates that the neuroinflammatory component of Parkinson’s disease can modulate adult neurogenesis, exerting both protective and harmful effects [110,111,112,113]. Recent findings indicate that IL-1β and IL-6 may be involved in astroglial modulation of adult neurogenesis and dopaminergic vulnerability, as suggested by elevated levels of these molecules in the cerebrospinal fluid of patients with PD [110,114] and as demonstrated by reactive astrocyte expression of IL-6 in the postmortem midbrain and iPSC-derived astrocytes [66,110,115]. Furthermore, some activated microglia populations can stimulate neuronal renewal in the adult CNS. Microglia pretreated with IL-4 or IFN-γ promote neurogenesis and oligodendrogenesis in SVZ-derived NPCs [116], while microglia pretreated with LPS inhibit both processes, confirming that an LPS-type inflammatory environment can block adult neurogenesis [117,118]. These findings underscore the importance of the inflammatory microenvironment in modulating adult neurogenesis, while the central role of astrocytes in this process is now widely recognized. Regarding astrocytes and DAergic neurodevelopment, the impact of glia–neuron crosstalk on the survival, proliferation, and differentiation of DA neurons is well recognized. Precursors of adult SVZ grown on monolayers of type 1 astrocytes show extensive neurogenesis [116,119], and multipotent neural stem cells in the midbrain and hindbrain of adult mice possess functional and dopaminergic neurogenic potential [120,121]. Preclinical studies also confirm the neurogenic capacity of mesencephalic progenitors isolated from adult mice [103,116]. Among the various experimental paradigms, only the direct co-culture of mesencephalic astrocytes with NPCs from the same region induces the dopaminergic phenotype, indicating a key role of astroglial factors in the induction of this phenotype [90,122]. MPTP damage and some pro-inflammatory chemokines, such as CCL2, CCL3, and CXCL10, stimulate Wnt1 expression in VM astrocytes, highlighting both regional specificity and the importance of a defined inflammatory microenvironment [90]. Wnt/β-catenin signaling is essential for activating adult neurogenesis in vitro and in vivo [123,124], and recombinant Wnt1 can promote neurogenesis in mesencephalic aNPCs. Overall, adult midbrain astrocytes, under specific neuroinflammatory conditions, can re-express regional factors such as Wnt1, contributing to homeostasis of damaged dopaminergic neurons, reduction of neuronal death, and survival, proliferation, and differentiation of dopaminergic progenitors. Thus, reactive VM astrocytes and Wnt/β-catenin signaling emerge as key mediators of MPTP-induced dopaminergic neuroplasticity [26,90] (Figure 4).

## 5. The Role of Microglial Cells in Parkinson’s Disease

As mentioned before, microglial cells, the main category of macrophages in the CNS parenchyma, express different receptors that are able to recognize both exogenous and endogenous CNS insults and are able to induce an immune response, showing an immune cell role [27]. These cells are essential in the immune surveillance of the presynaptic microenvironment and play a key role in the synaptic remodeling by regulating proteolytic and phagocytic processes that are fundamental in axonal and dendritic terminals pruning. Moreover, microglia exhibit the capacity to recruit astrocytic cells, or may be recruited by the latter, showing physiological properties needed for synaptic transmission and modulation of neural and synaptic plasticity, since microglia cells are able to promote the remotion of non-essential synapses, thus increasing neuronal transmission and synapse remodeling [120,121,122,123].

In general, microglial cells sustain brain homeostasis and tissue repair by stimulating phagocytic clearance and providing trophic support. However, conditions of homeostasis loss or tissue changes induce cellular morphology and surface phenotype modifications and increased proliferative responses leading to the “activated state” of microglia [124]. The activation of microglia, one of the hallmarks of the neuroinflammatory process, is a typical pathophysiological feature of neurodegenerative disorders, including PD, suggesting that it can influence the neuron-to-neuron transmission of α-synuclein [35,125,126]. Specifically, microglial activation in the CNS shows a high degree of heterogeneity, functionally categorized by distinct pro-inflammatory and anti-inflammatory phenotypes, each characterized by specific markers and released mediators generating either cytotoxic or neuroprotective effects [35,127]. In the past, the “M1 vs. M2” paradigm originating from macrophage immunology has been employed for an extended period to characterize microglial activity in the central nervous system (CNS). In this classification, microglial cells “M1” are considered as pro-inflammatory phenotypes, with the production of cytokines and toxic mediators (e.g., NO and ROS), while the “M2” would execute anti-inflammatory and tissue repair tasks. Lately, the research has questioned the M1/M2 dichotomy, as it does not reflect the full complexity of the microglial compartment in vivo [128]. A recent study has illustrated how microglial profiles in humans are dispersed along complicated spatiotemporal gradients, emphasizing various subgroups associated with development, brain area, and illness status [129]. This heterogeneity has also been demonstrated in Alzheimer’s disease, where He and colleagues [130] discuss how microglia play both protective and harmful roles with dynamics that cannot be explained by M1 vs. M2 classification alone. Recent studies using single-cell RNA-seq (scRNA-seq) and spatial transcriptomics techniques have shown that after brain insults (ischemia, neuroinflammation, or chronic damage), multiple microglial subpopulations emerge, each with distinct transcriptional profiles and often not traceable to the “classical” M1 or M2 markers [131]. For example, in the study of Ma and colleagues [132], 37,614 microglial cells from a mouse model of cerebral ischemia were investigated, and eight different subpopulations were found using scRNA-seq. Among them, some presented features “M1-like” after ischemia, but others showed profiles with little inflammation, and none obviously “M2-like”, suggesting that the post-ischemia microglial response is not limited to a straightforward M1 → M2 transition. It has also been found that throughout development, aging, disease, and pathological situations, microglial cells assume dynamic and environmentally controlled states, with transcriptional profiles modified by specific enhancers and transcription factors like SALL1 and SMADs [133]. This means that microglia can activate programs that combine pro-inflammatory but also neuroprotective elements not necessarily attributable to the classic M1 or M2 profile [128]. All of this can lead to misleading interpretations in in vivo models and, thus, the M1 vs. M2 paradigm is now obsolete. In its place, the concept of “highly heterogeneous, dynamic, and contextual microglial population” emerges, which, in response to different stimuli, can assume different functional states [134].

In neuroinflammatory disease, the pro-inflammatory phenotype produces cytokines and NO, decreasing the release of neurotrophic factors and exacerbating the inflammatory and cytotoxic process. Conversely, the anti-inflammatory phenotype displays anti-inflammatory cytokines and increased neurotrophic factors expression, playing a role in downregulation, protection, or repair to counteract the inflammation (Figure 5) [27]. The dynamic changes in the different phenotypes are critically associated with neurodegenerative diseases, and several studies suggest that shifting and/or modulation of microglia polarization from the pro-inflammatory to the anti-inflammatory phenotype, through important signal pathways, could be essential in neurodegenerative disorders, including PD [135]. In the early stage of PD, microglia cells are activated by α-syn, pathogens, or environmental toxins, maintaining a “resting” state and are not correlated with clinical severity, but in a state of continued microglial activation, harmful events taking place, and the aggregation of α-syn could directly induce microglia toward the M1 pro-inflammatory phenotype, exacerbating motor symptoms and increasing the neuronal damage to neighboring neurons [127,135,136].

Increasing scientific evidence obtained from both preclinical and human studies suggested that microglia activation occurs in the PD preclinical stages, even before the α-synuclein aggregation and DAergic neurons degeneration, suggesting that the activated microglia contribute to the pathological α-synuclein aggregation.

In PD, the activated microglia reduce α-synuclein degradation and promote its misfolding and aggregation. In addition, misfolded α-synucleins species, in turn, activate the inflammatory response of microglia and promote microglial migration, thus forming a positive feed-forward loop [125]. Pathogenic molecules, such as LPS and/or IFN, or protein aggregates, including α-synuclein, stimulate the pro-inflammatory M1 phenotype, which in turn increases the release of pro-inflammatory mediators, such as ROS, IL-1β, iNOS, and TNFα, resulting in increased neuronal damage and affecting synaptic function and mitochondrial homeostasis [35]. Moreover, the release of aggregated α-syn from affected neurons induces a dysregulation of the microglia immune response by blocking the phagocytosis process mediated by microglial cells, resulting in an impairment of the clearance of accumulated proteins or cellular debris, a mechanism that contributes to neurodegeneration observed in patients with PD [35,137]. In this context, several studies have demonstrated that signal transducer and activator of transcription 1 (STAT1) is involved in M1/M2 polarization, and the activation of the JAK2/STAT1 pathway enhances the macrophage polarization into the M1 phenotype, leading to a harmful inflammatory event in the hippocampus [138,139,140,141].

IFN-γ and IL-6, two of the most potent activators of the JAK/STAT pathway, are elevated in PD, and evidence suggests that this pathway could play a role in the development of PD by modifying microglial polarization, supporting the hypothesis to explore the role of JAK inhibitors to improve neuroinflammation in PD [35,142,143]. Preclinical and human studies have suggested the key role of microglia-mediated neuroinflammation in PD [35] that could be a target for the development of new therapies, taking into consideration not only the molecular targets of the disease, but also the role played by microglia.

## 6. Glial-Targeted Therapies: Strengths and Limitations

Traditionally, research and development of therapies for neurological diseases have focused predominantly on neurons. However, it is now widely recognized that glial cells—astrocytes, oligodendrocytes, and microglia—are not simple supporting cells, but dynamic and crucial players that actively modulate neuronal function, synaptic plasticity, CNS homeostasis, and exert a role in responses to injury and disease. As a result, glial target therapies have emerged as a new and promising strategy for the treatment of a wide range of neurological disorders, from neurodegenerative diseases and spinal injuries to psychiatric disorders [144].

Phagocytic “cleansing” by glia is a critical target: microglia and astrocytes can remove non-essential synapses, cellular debris, and toxic proteins (such as α-synuclein), but a dysregulation in the endolysosome process or genetic alterations related to PD can impair this function [145]. Restoring or enhancing this phagocytic capacity could be therapeutically useful for reducing pathological protein accumulation and neurodegeneration. Recent studies have demonstrated that CB2 receptor activation in astrocytes can promote autophagy and promote FoxG1-mediated degradation of the NLRP3 inflammasome, reducing inflammation in mouse models of PD [146]. Some therapies have aimed to restore a protective phenotype “by moving” microglia from a pro-inflammatory to an anti-inflammatory/reparative state, modulating the release of neurotoxic cytokines and chemokines, promoting neuroprotection by stimulating cells to release neurotrophic factors (e.g., BDNF or GDNF) and antioxidant factors that support neuronal survival and their function.

In a study using the MPTP model, the authors highlighted that microglia and astrocytes change their phenotype (inflammatory or phagocytic) over time and suggested that this phenomenon could be related to the brain region considered, assuming that the “therapeutic window” able to modulate these cells could change during the state of the disease [146].

Another important therapeutic strand concerns iron metabolism and ferroptosis: according to a recent review, glial cells (microglia, astrocytes, and oligodendrocytes) contribute to neuronal death by regulating iron homeostasis and lipid peroxidation, effectively modulating ferroptosis in DAergic neurons [147].

Considering that all strategies ultimately aim to preserve neuronal vulnerability, targeting glial cells by modulating inflammation or autophagy could represent a targeted therapeutic approach that could potentially offer more specific and long-lasting treatments for neuroinflammatory diseases, including Parkinson’s disease (Figure 6).

Furthermore, from a therapeutic and translational perspective, the new microglial phenotypic view suggests targeting specific functional sub-populations by modulating metabolic function or transcriptional/regulatory factors that improve microglial status in a given context [133]. These observations are complemented by proposals for novel biomarkers and therapeutic strategies that exploit specific microglial pathways [148]. New methodological and applied approaches are further expanding the field by linking morphology and function and demonstrating how nanomaterials can selectively modulate different microglial states [149,150]. Overall, the literature agrees that M1 vs. M2 remains a simplified model useful only in specific contexts, while microglia, in vivo, emerge as a highly plastic and diverse population, best described through transcriptomic atlases, intermediate states, and dynamic functional trajectories. To improve our understanding and therapeutic interventions on the microglial system, it is, therefore, essential to characterize real states that reflect the biological complexity of organisms and thus be able to develop interventions targeted to specific subpopulations or regulatory mechanisms (transcription factors, metabolism, and signaling).

Despite the therapeutic potential provided by astrocytes and microglia in PD, significant difficulties should be considered: firstly, cellular heterogeneity and timing, because glia can assume pro-inflammatory, neuroprotective, or senescent states, which change over time and between brain regions, and therefore, with a non-specific target, could compromise the beneficial sub-populations [35]. Another limitation is BBB permeability, which limits the safe and targeted delivery of glial-based drugs, with the risk of peripheral off-target effects [151]. Furthermore, the targeting of chronic immune modulation shows several difficulties, including that the inflammasome inhibitors, such as NLRP3, must be carefully evaluated from the perspective of safety and the therapeutic window [148]. Finally, translation from preclinical models to humans is problematic: many mice or organoid models are not able to replicate the age of onset and/or development of the disease, as well as the complexity of immunity or heterogeneity of human glia, limiting the predictivity of the results [152]. To overcome these limitations, future strategies include spatiotemporal targeting with cell-type-specific viral or non-viral vectors (e.g., astrocyte promoters), which allow for the expression of neuroprotective factors locally; functional reprogramming of glia using small molecules or biologics that promote metabolic or antioxidant-supporting phenotypes; the development of selective inflammasome inhibitors with controlled CNS release; the use of advanced delivery technologies, such as Trojan-horse nanoparticles or intracerebral delivery to bypass the BBB; and the generation of more predictive human models, such as iPSC-derived glia, organoids, or single-cell approaches to identify translational biomarkers [35,152].

## 7. iPSC-Derived Astrocytes as a Model to Study the Role of the Wnt/β-Catenin Pathway in Parkinson’s

During recent years, astrocytes derived from induced pluripotent stem cells (iPSCs) have provided a unique experimental model to study the mechanisms involved in PD. There are many studies demonstrating how iPSC astrocytes obtained from patients with LRRK2 or GBA mutations, for example, exhibit different phenotypes involving metabolic dysfunction, abnormal inflammatory responses, α-synuclein accumulation, and reduced antioxidant capacity. This provided a very important human model for understanding how glia contribute to neuronal vulnerability [66,119,153], thus reinforcing the idea of what has already been demonstrated in MPTP mouse models on the survival and regeneration of dopaminergic neurons. de Rus Jacquet and colleagues [154] developed an innovative on-chip blood–brain barrier model by integrating iPSC astrocytes carrying the LRRK2 G2019S mutation, demonstrating that these cells contribute to greater barrier dysfunction and aberrant inflammatory responses, highlighting the pathogenetic role of astroglial alterations in PD. In a complementary approach, Song et al. [155] showed that cografting midbrain astrocytes together with dopaminergic neurons improves neuronal transplant engraftment, survival, and functionality, indicating how astrocytes can act as true co-therapeutics, and opening concrete perspectives for the use of iPSC-derived astrocytes in cell replacement protocols. Finally, Szeky et al. [156] described an efficient and reproducible protocol for deriving mature astrocytes from human induced pluripotent stem cells (hiPSCs) characterized by key functional properties, such as glutamate uptake and calcium response, thus providing a solid technical basis for the standardized employment of these cells in co-cultures and in experimental models of neurodegeneration. The combination of these data offers the possibility of investigating whether and how astrocytes derived from human iPSCs can reproduce the same protective mechanisms, allowing the modulation of Wnt/Nrf2 to be directly tested in co-cultures with dopaminergic neurons derived from iPSCs [156,157,158]. This approach provides a conceptual and experimental bridge between in vivo observations in the MPTP model and human pathology, with important perspectives for the development of glia-targeted therapies.

## 8. Conclusions

In summary, astrocytes possess adaptive and specific mechanisms that can be activated in response to brain damage, as observed in the pathogenesis of Parkinson’s disease (PD). Although bidirectional communication between dopaminergic neurons and glia is crucial for maintaining homeostasis in the midbrain, chronic neurodegeneration conditions—characterized by excessive oxidative and nitrosative stress, as well as aberrant production of cytotoxic mediators—can induce astrocytes and microglia to become dysfunctional. In this state, these cells lose the effectiveness necessary to protect nigral DAergic neurons or to promote neuronal recovery [26,36,44,159]. Indeed, it is well known that both astrocytes and microglial cells can protect neurons and promote neurogenesis, while during a chronic inflammatory process their activation contributes to the neurodegenerative process and to DAergic neurons dysfunction.

Considering the primary role of astrocytes and neurons in the functions and development of the CNS, the alteration of this communication, induced by genetic and/or environmental factors, can significantly increase the vulnerability of the nigrostriatal DAergic system [160,161,162,163]. The identification and/or development of safe and effective therapies requires joint action: on the one hand, inhibiting the release of harmful factors by lesioned astrocytes; on the other hand, replacing these cells (both in modified and unmodified form in vitro), supplementing the treatment with a combination of neurotrophins and antioxidants [164,165]. Knowledge of how the complex DAergic recovery mechanisms are altered in PD provides the basis for the identification of new regenerative therapies. Among these, the intervention on the Wnt pathway, thanks to its ability to amplify the endogenous regenerative potential of DAergic neurons, is configured as a strategy with profound implications for the treatment of PD. We know that in neurodegenerative diseases, downregulation of this system contributes to increased Aβ production, tau hyperphosphorylation, synapse loss, neurodegeneration, and BBB damage [166]. Controlled reactivation of this pathway, for example, by modulating the LRP6 co-receptor, inhibiting the GSK-3β kinase to stabilize β-catenin, modulating gene transcription of protective factors promoting neurogenesis, may be an important therapeutic avenue [167]. On the other hand, the literature suggests that Wnt activation is “context-dependent”; indeed, inappropriate activation can lead to aberrant proliferation or negative consequences [168]. Overall, Wnt/β-catenin, when adequately modulated, represents a promising and rational target for neuroprotective treatments in neurodegenerative diseases.

In conclusion, therapies aimed at improving the efficiency of neurogenesis, the functional integration of newly produced neurons, or those that act directly on astrocytes to promote in situ recovery in the nigrostriate, represent promising new targets to explore in PD experimental models. Astrocyte-focused therapies offer great hope for shifting the balance towards neuroprotection and recovery in experimental models and, potentially, in clinical settings [26,90]. Moreover, microglial cells and astrocytes, working together, can also eliminate extracellular α-synuclein, exerting a protective effect on neurons [35]. Therefore, targeting astrocyte- and microglia-specific protective mechanisms could be considered as a potential therapeutic strategy to counteract DAergic neurodegeneration, and at the same time, functional reprogramming of microglia polarization into its different phenotypes could be investigated as a neuroprotective tool against neuroinflammation-induced dopaminergic neuronal degeneration in SN in PD.

## Figures and Tables

**Figure 1 ijms-26-11880-f001:**
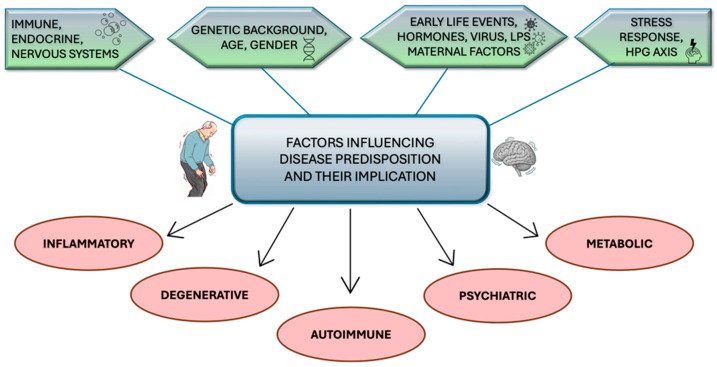
Perinatal factors shaping glial inflammation and adult susceptibility to inflammatory diseases. Genetic factors can interact with maternal hormonal influences and with exogenous agents to which the mother and fetus are exposed, modifying the development of the neuroendocrine-immune system, and mainly affecting glial cells, altering the mechanisms of the brain response to inflammation, thus contributing to individual vulnerability, susceptibility, and predisposition to inflammatory, autoimmune, and psychiatric disorders.

**Figure 2 ijms-26-11880-f002:**
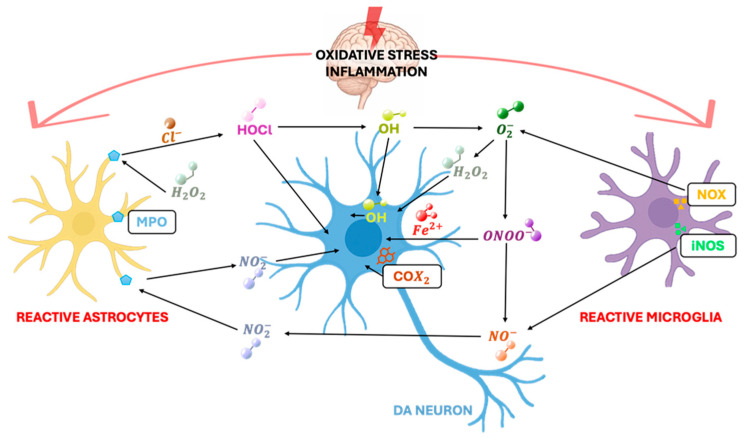
An illustrative diagram of the mechanisms through which inflammatory and oxidative stress mediators contribute to the death of dopaminergic neurons. Exposure to MPTP induces oxidative stress, mitochondrial dysfunction, and alterations in astrocyte function, events that lead to microglial activation. The latter increases the expression of iNOS and NADPH oxidase, favoring the concomitant production of nitric oxide and superoxide. The interaction of these reactive species leads to the formation of peroxynitrite (ONOO-), a potent toxic agent capable of nitrating tyrosine and generating hydroxyl radicals. Reactive astrocytes, moreover, can produce high amounts of myeloperoxidase (MPO) that converts hydrogen peroxide and chloride anion into hypochlorous acid (HOCl), a highly reactive molecule able to induce macromolecular damage and amplify oxidative damage. Dopaminergic neurons also contribute to oxidative stress associated with inflammation through increased COX-2 expression.

**Figure 3 ijms-26-11880-f003:**
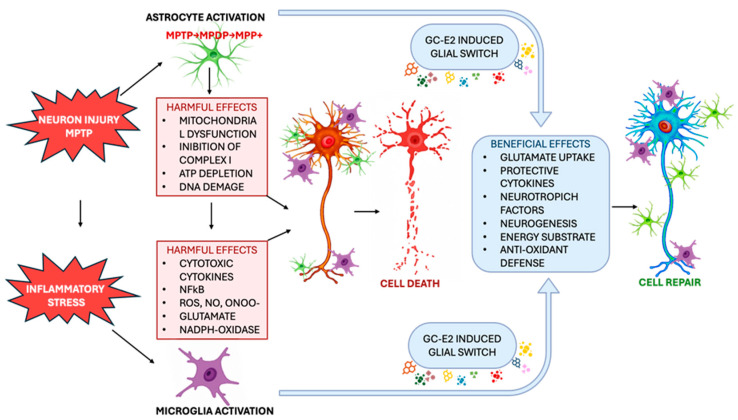
The pattern of harmful and protective glial pathways in Parkinson’s disease. Chemical insults (for example, MPTP) and/or genetic and environmental factors damage nigrostriatal dopaminergic neurons, activating astrocytes and microglia. MPTP is converted to MPP+ by astrocytes and accumulates in dopaminergic neurons, while microglial cytotoxic mediators amplify inflammation and oxidative stress. Astrocytes and microglia can instead protect neurons by removing free radicals and glutamate, providing energy support and trophic factors, and promoting repair and neurogenesis. In conditions of chronic inflammation, however, glial cells become release cytotoxic mediators, contributing to the death of dopaminergic neurons.

**Figure 4 ijms-26-11880-f004:**
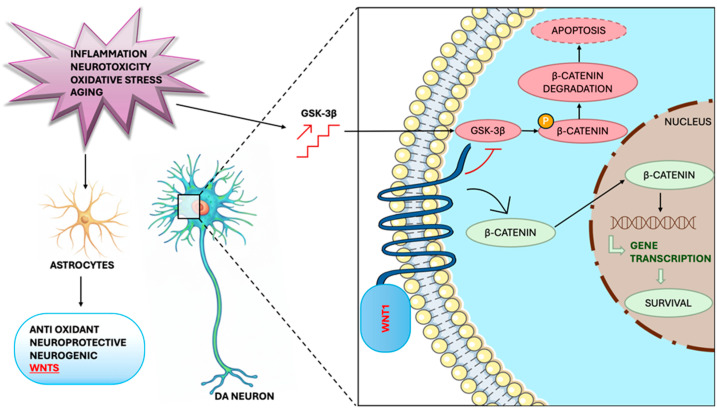
Reactive astrocytes and the Wnt pathway in a MPTP mouse model of PD. Astrocyte-derived Wnts, particularly Wnt1, protect the integrity of dopaminergic neurons through blocking GSK-3β-induced phosphorylation and proteosomal degradation in the neuronal β-catenin pool. Activation of Wnt/β-catenin signaling can also promote neurogenesis from adult midbrain progenitors. Neurotoxic damage or increased oxidative load due to aging can antagonize Wnt/β-catenin signaling in DAergic neurons, activating GSK-3β and consequently degrading β-catenin, making dopaminergic neurons vulnerable. Neuronal damage also causes astrocytes to activate and produce various neurotrophic and growth factors that protect DAergic neurons by promoting neurogenesis and cell survival.

**Figure 5 ijms-26-11880-f005:**
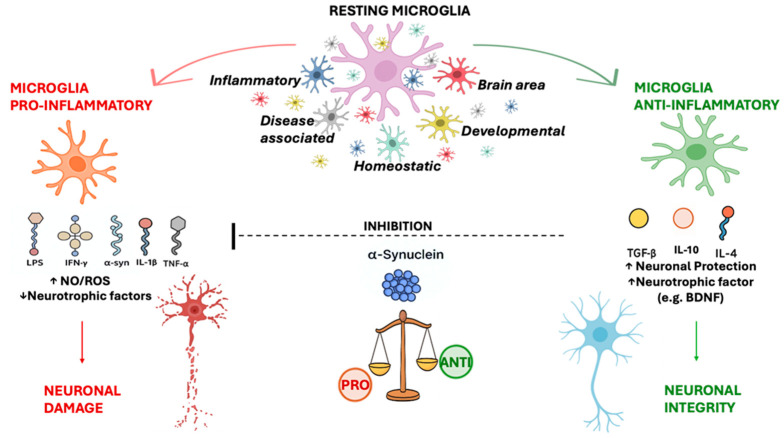
The role of microglial cells. In response to local stimuli (e.g, α-syn), microglia polarize from a “resting” state to “activated” microglia. The pro-inflammatory phenotype exhibits pro-inflammatory cytokines (e.g., IL-1β and TNF-α), decreasing the release of neurotrophic factors and exacerbating the inflammatory process. The anti-inflammatory phenotype displays anti-inflammatory cytokines (e.g., TGF-β1, IL-10, and IL-4) and increased neurotrophic factors (e.g., BDNF) expression, playing a role in neuronal protection. Moreover, α-syn aggregation induces microglia toward the pro-inflammatory phenotype, exacerbating the neuroinflammatory process in PD. The anti-inflammatory phenotype can also inhibit alpha-synuclein toxicity by promoting its elimination and releasing factors that reduce inflammation and neuronal stress.

**Figure 6 ijms-26-11880-f006:**
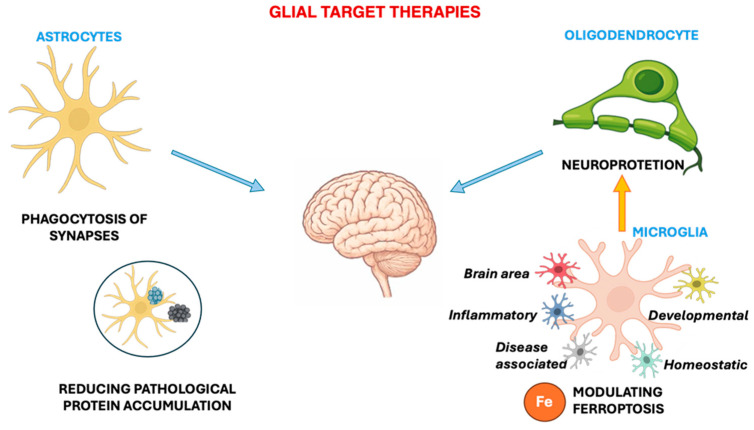
Glial therapy strategies. The phagocytic capacity of glial cells (e.g., astrocytes) could be therapeutically useful for reducing pathological protein accumulation (e.g., α-syn). Moreover, a therapeutic strategy could target the restoration of the balance between the pro- and anti-inflammatory microglial phenotypes, exerting a neuroprotective activity. Glial cells (microglia, astrocytes, and oligodendrocytes) modulating ferroptosis in DAergic neurons could be a target for neuroprotection.

## Data Availability

No new data were created or analyzed in this study. Data sharing is not applicable to this article.

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
