# Peer review of "Role of Reactive Astrocytes and Microglia: Wnt/β-Catenin Signaling in Neuroprotection and Repair in Parkinson’s Disease"

_ijms, 2025, doi:10.3390/ijms262411880_

Round 1

Reviewer 1 Report

Comments and Suggestions for Authors

In this review, the authors examine the current state of knowledge regarding the role of astrocytes in dopaminergic neurodegeneration, neuroprotection, and neurorepair. The review focuses on the relationship between astrocytic origin factors and neurogenic signals that mediate MPTP-induced plasticity in dopaminergic neurons of the nigrostriatal system.

This is a well-presented review that addresses a very interesting topic. The review appears solid and comprehensive and leads to consistent conclusions. However, some aspects require clarification, further explanation, or correction. The main issue is that the review partially overlooks the relevant role of microglia, in addition to astrocytes, in the pathophysiology of Parkinson’s disease and as essential cells involved in the Wnt/β-catenin pathway in Parkinson’s disease.

Comments:

  1. As in previous reviews on this topic, the authors focus primarily on astrocytes; however, microglial cells are also significantly involved, both in the chronic inflammatory process and in the pathology itself. The authors frequently mention microglial cells as relevant participants. Therefore, it would be appropriate to include microglia in the title, alongside astrocytes, and to add a section similar to section 3, but devoted to microglia, which is also discussed in sections 2 and 4.
  2. Are there studies focused on microglia that could also be included in section 5, such as “glial target therapies”?
  3. In the conclusion section, or before it, it would be useful to include a brief section on future perspectives for both astrocytes and microglia. This is already partially addressed in the conclusion, but it could be further emphasized, considering possible limitations and troubleshooting.
  4. Consider including a figure focused on the role of microglia. Microglia are already included in figure 3.

MINOR
1. Line 398, Figure 4: Correct the typesetting of "astrocytes."

  1. Figure 2: The arrows are confusing; they would be clearer if they were black with solid lines.
  2. The text in the figures should be large enough to be readable.

Author Response

1. As in previous reviews on this topic, the authors focus primarily on astrocytes; however, microglial cells are also significantly involved, both in the chronic inflammatory process and in the pathology itself. The authors frequently mention microglial cells as relevant participants. Therefore, it would be appropriate to include microglia in the title, alongside astrocytes, and to add a section similar to section 3, but devoted to microglia, which is also discussed in sections 2 and 4.

A: We thank Reviewer #1 for carefully reading and analyzing the manuscript and for her/his supportive comments. As suggested by Reviewer #1 we added information about the role of microglia in the title and in the keywords. Moreover, we added a section where we discussed the role of microglia (new section 5 in the revised version of the manuscript). This is now stated in the revised version of the manuscript (Pages 12-13 – Lines 471-536).

2. Are there studies focused on microglia that could also be included in section 5, such as “glial target therapies”?

A: We thank Reviewer #1 for this comment that helped us to significantly improve our manuscript. We added a new section where we discussed this topic focusing on strengths and limitations (new section 6 in the revised version of the manuscript). This is now stated in the revised version of the manuscript (Pages 13-14 – Lines 549-604). Moreover, we added a new Figure where we summarized the results showed in the literature about glial target therapies in neurodegenerative diseases with a particular focus on Parkinson’s Disease (see new Figure 6 in the revised version of the manuscript).

3. In the conclusion section, or before it, it would be useful to include a brief section on future perspectives for both astrocytes and microglia. This is already partially addressed in the conclusion, but it could be further emphasized, considering possible limitations and troubleshooting.

A: We added a new section where we discussed this topic focusing on strengths and limitations (new section 6 and new Figure 6 in the revised version of the manuscript). The role of both astrocytes and microglia is also discussed in the Conclusion section (Page 16 – Lines 652-655, Lines 673-679).

4. Consider including a figure focused on the role of microglia. Microglia are already included in figure 3.

A: We thank Reviewer #1 for his/her suggestion. We included a Figure focusing on the role of microglia (see new Figure 5 in the revised version of the manuscript).

MINOR
1. Line 398, Figure 4: Correct the typesetting of "astrocytes.

A: We corrected the mistakes. All changes are highlighted in yellow in the revised version of the manuscript.

2. Figure 2: The arrows are confusing; they would be clearer if they were black with solid lines.

A: We thank Reviewer #1 for his/her suggestion. As suggested by Reviewer #1 we changed the format of Figure 2 and also for the other Figures in order to be clearer (arrows black with solid lines 1.5 pt).  

3. The text in the figures should be large enough to be readable.

A: We thank Reviewer #1 for his/her suggestion. As suggested by Reviewer #1 we changed the text in all Figure using a 14-18 point size.

Reviewer 2 Report

Comments and Suggestions for Authors

The review “ROLE OF REACTIVE ASTROCYTES AND WNT/β-CATENIN 2 SIGNALING IN NEUROPROTECTION AND REPAIR IN 3 PARKINSON’S DISEASE” by Dr. Grasso and colleagues is very interesting. It addresses the highly significant topic of Parkinson's disease pathogenesis and symptomatology. Its primary strength lies in challenging the predominant neuronal perspective by highlighting the potential role of glial cells in both pathobiology and compensatory mechanisms. By proposing this shift in focus, the review promotes valuable new experimental hypotheses for a condition that impacts a significant segment of the aging population. However, the paper is not in its mature form yet. There are numeros errors, sentences’ patchworking and illogical jumps.

The authors should expand the discussion to include the potential for microglial inactivation, as they introduce the concept of microglial activation. Furthermore, positioning caffeine and tobacco as neuroprotective factors is unconventional; the logic and evidence supporting this claim require careful re-examination.
The references are appropriate and current, and the schematic figures are well-suited for a review article.

There the following points to correct:
-    Line 16, MP – abbreviation needs to be deciphered
-    Line 60 and 61 – check the meaning, is ts right?  Caffeine and tobacco consumption are now considered as neuroprotection? “instead, they seem to reduce the risk, the consumption of caffeine and tobacco, a healthy and balanced diet, movement and social relationships [9].”
-    Line 75, typo: “deurodegeneration”
-    Line 81, typo: “Astrocyes”
-    Line 82, typo: “neurodegerative”
-    Lines 110-112 “Once activated, microglia display conspicuous functional plasticity and ultimately transform into a macrophage-like phenotype”.. The authors should describe the next stages of activated microglial cells – do they always behave as macrophage, or there are conditions to return to a normalized state
-    Line 153, typo: ” E’ this compound”
-    Line 154, typo: “domaminergic neurons”
-    Line 202, typo: ” to oligodendricites”
-    Line 224, typo: “monoamine oxidai - B (MAO-B)”
-    Line 253, word missing :” In mice lacking, impairment of endogenous..”
-    Line 292, typo: “daergic neurons”
-    Lines 292-297: the sentence is unclear and needs re-writing
-    From Lines 309 and below: the Wnt family needs to be described better
-    Line 409: the abbreviation “hiPSCs” is not deciphered anywhere
-    Conclusion -it would be logical to say smth on microglial cells, not only about the astrocytes

Comments on the Quality of English Language

too many typos

Author Response

The review “ROLE OF REACTIVE ASTROCYTES AND WNT/β-CATENIN 2 SIGNALING IN NEUROPROTECTION AND REPAIR IN 3 PARKINSON’S DISEASE” by Dr. Grasso and colleagues is very interesting. It addresses the highly significant topic of Parkinson's disease pathogenesis and symptomatology. Its primary strength lies in challenging the predominant neuronal perspective by highlighting the potential role of glial cells in both pathobiology and compensatory mechanisms. By proposing this shift in focus, the review promotes valuable new experimental hypotheses for a condition that impacts a significant segment of the aging population. However, the paper is not in its mature form yet. There are numeros errors, sentences’ patchworking and illogical jumps.

The authors should expand the discussion to include the potential for microglial inactivation, as they introduce the concept of microglial activation. Furthermore, positioning caffeine and tobacco as neuroprotective factors is unconventional; the logic and evidence supporting this claim require careful re-examination.
The references are appropriate and current, and the schematic figures are well-suited for a review article.

A: We thank Reviewer #2 for carefully reading and analyzing the manuscript and for her/his supportive comments. As suggested by the Reviewer, we added information about the role of microglia and we added a new section where we discussed the role of microglia (new section 5 in the revised version of the manuscript). This is now stated in the revised version of the manuscript (Pages 12-13 – Lines 473-536). Furthermore, we re-examined the evidence about the role of caffeine and tobacco (Pages 2-3 – Lines 62-95 in the revised version of the manuscript).

There the following points to correct:
-    Line 16, MP – abbreviation needs to be deciphered
-    Line 60 and 61 – check the meaning, is ts right?  Caffeine and tobacco consumption are now considered as neuroprotection? “instead, they seem to reduce the risk, the consumption of caffeine and tobacco, a healthy and balanced diet, movement and social relationships [9].”
-    Line 75, typo: “deurodegeneration”
-    Line 81, typo: “Astrocyes”
-    Line 82, typo: “neurodegerative”
-    Lines 110-112 “Once activated, microglia display conspicuous functional plasticity and ultimately transform into a macrophage-like phenotype”.. The authors should describe the next stages of activated microglial cells – do they always behave as macrophage, or there are conditions to return to a normalized state
-    Line 153, typo: ” E’ this compound”
-    Line 154, typo: “domaminergic neurons”
-    Line 202, typo: ” to oligodendricites”
-    Line 224, typo: “monoamine oxidai - B (MAO-B)”
-    Line 253, word missing :” In mice lacking, impairment of endogenous..”
-    Line 292, typo: “daergic neurons”
-    Lines 292-297: the sentence is unclear and needs re-writing
-    From Lines 309 and below: the Wnt family needs to be described better
-    Line 409: the abbreviation “hiPSCs” is not deciphered anywhere
-    Conclusion -it would be logical to say smth on microglial cells, not only about the astrocytes

A: We thank Reviewer #2 for his/her suggestion. We rephrased several sentences in the revised version of the manuscript, and all changes are highlighted in yellow. Moreover, as suggested by Reviewer #2, we described the stage of activated microglial cells (Page 4, Lines 147-154). As suggested by Reviewer #2, we discussed the role of microglia in the Conclusion section (Page 16 – Lines 652-655, Lines 673-679). Finally we added a paragraph were we discussed the Wnt pathway (Lines 357-375 in the revised version of the manuscript).

Round 2

Reviewer 1 Report

Comments and Suggestions for Authors

The authors have conveniently addressed the questions and comments, consequently the manuscript has been improved in precision. 

Author Response

Thank you for taking the time to review my manuscript